# Transcriptome Sequencing Reveals Autophagy Networks in Rat Livers during the Development of NAFLD and Identifies Autophagy Hub Genes

**DOI:** 10.3390/ijms24076437

**Published:** 2023-03-29

**Authors:** Jian Xie, Qiuyi Chen, Yongxia Zhao, Mingxia Luo, Xin Zeng, Lin Qin, Daopeng Tan, Yuqi He

**Affiliations:** 1Guizhou Engineering Research Center of Industrial Key-Technology for Dendrobium Nobile, Zunyi Medical University, Zunyi 563000, China; xiejian@zmu.edu.cn (J.X.); chenqiuyi@zmu.edu.cn (Q.C.); zhaoyongxia@zmu.edu.cn (Y.Z.); luomingxia@zmu.edu.cn (M.L.);; 2Key Laboratory of Basic Pharmacology of Ministry of Education and Joint International Research Laboratory of Ethnomedicine of Ministry of Education, Zunyi Medical University, Zunyi 563000, China; 32011 Cooperative Inovational Center for Guizhou Traditional Chinese Medicine and Ethnic Medicine, Zunyi Medical University, Zunyi 563000, China; 4Department of Medical Genetics, Zunyi Medical University, Zunyi 563000, China; zengxin@zmu.edu.cn

**Keywords:** autophagy, NAFLD, high-fat diet, transcriptome, biomarkers

## Abstract

(1) Autophagy is an important biological process in cells and is closely associated with the development and progression of non-alcoholic fatty liver disease (NAFLD). Therefore, this study aims to investigate the biological function of the autophagy hub genes, which could be used as a potential therapeutic target and diagnostic markers for NAFLD. (2) Male C57BL/6J mice were sacrificed after 16 and 38 weeks of a high-fat diet, serum biochemical indexes were detected, and liver lobules were collected for pathological observation and transcriptome sequencing. The R software was used to identify differentially expressed autophagy genes (DEGs) from the transcriptome sequencing data of mice fed with a normal diet for 38 weeks (ND38) and a high-fat diet for 38 weeks (HFD38). Gene ontology (GO) and the Kyoto Encyclopedia of Genes and Genomes (KEGG) analysis were performed on the DEGs, a protein–protein interaction (PPI) network of the DEGs was established using the STRING data website, and the results were visualized through Cytoscape. (3) After 16 weeks and 38 weeks of a high-fat diet, there was a significant increase in body weight, serum total cholesterol (TC), low-density lipoprotein-cholesterol (LDL-C) and triglycerides (TG) in mice, along with lipid accumulation in the liver, which was more severe at 38 weeks than at 16 weeks. The transcriptome data showed significant changes in the expression profile of autophagy genes in the livers of NAFLD mice following a long-term high-fat diet. Among the 31 differentially expressed autophagy-related genes, 13 were upregulated and 18 were downregulated. GO and KEGG pathway analysis revealed that these DEGs were primarily involved in autophagy, cholesterol transport, triglyceride metabolism, apoptosis, the FoxO signaling pathway, the p53 signaling pathway and the IL-17 signaling pathway. Four hub genes were identified by the PPI network analysis, of which *Irs2, Pnpla2* and *Plin2* were significantly downregulated, while *Srebf2* was significantly upregulated by the 38-week high-fat diet. (4) The hub genes *Irs2*, *Pnpla2*, *Srebf2* and *Plin2* may serve as key therapeutic targets and early diagnostic markers in the progression of NAFLD.

## 1. Introduction

Nonalcoholic fatty liver disease (NAFLD) is a chronic liver disease that has become a major threat to human health worldwide due to factors such as excessive fat accumulation, intestinal ecological imbalance, insulin resistance (IR) [1], liver inflammation and gene effects [2,3]. NAFLD can progress to more severe conditions such as nonalcoholic steatohepatitis (NASH), cirrhosis, and even hepatocellular carcinoma (HCC) [4]. At present, liver biopsy is the gold standard for the diagnosis of NAFLD, but due to the limitations of invasive trauma, poor acceptability and high requirement for manipulation, this method is widely limited in clinical practice or trials. In recent years, noninvasive imaging diagnosis methods have been developed, while this method has a greater accuracy and repeatability in the diagnosis of NAFLD, and it also comes with higher economic costs and a risk of complications [5,6]. Therefore, there is a need to find new biomarkers and diagnostic methods to specifically identify NAFLD in vivo for a more accurate and preventive treatment of NASH and HCC.

Autophagy is a stable self-degradation mechanism in cells that can degrade dead organelles, misfolded proteins and other molecules within cells, leading to the reuse of intracellular substances and providing energy for cell survival [7]. Autophagy can be classified into three forms based on the different substrate transport mechanisms: microautophagy, macroautophagy and molecular chaperone-mediated autophagy [8]. It has been found that the autophagy of hepatocytes is closely related to obesity, NAFLD and other metabolic diseases [9], making it a significant potential mechanism for the development and progression of NAFLD [10]. Autophagy plays a vital role in regulating lipid metabolism, improving insulin resistance and reducing hepatocyte injury and inflammation through multiple pathways [11,12]. Research has shown that several genes, such as *SRY-box transcription factor 9 (Sox9), C-C motif chemokine ligand 20 (Ccl20), C-X-C motif chemokine ligand 1 (Cxcl1), the cluster of differentiation 24 (Cd24)* and *car-bohydrate sulfotransferase 4 (Chst4)* were involved in the aggravation of NAFLD, and their expression increased with disease progression [13]. In addition, Ma et al. identified several autophagy-related genes, including *nitric oxide synthase 3 (Nos3), insulin-like growth factor 1 (Igf1), vesicle-associated membrane protein 8(Vamp8), fos proto-oncogene (Fos)*, and *heme oxygenase 1 (Hmox1)*, in the GSE89632 data set and recognized the MAPK signaling pathway as an important pathway, with *Jun proto-oncogene (Jun)* being a key gene involved in NAFLD progression [14]. Nonetheless, limited research has been conducted on the differential autophagy-related biomarkers associated with disease progression and the potential functional pathways of NAFLD.

In this study, we established a high-fat diet mouse model to induce NAFLD in both short-term (16 weeks, HFD16) and long-term (38 weeks, HFD38) groups. By integrating transcriptome sequencing data with the literature and database searches, we collected a total of 786 autophagy-related genes and revealed the effects of a long-term high-fat diet on the liver autophagy gene network of NAFLD mice. Targeted autophagy core genes were screened at the transcriptome level.

## 2. Results

### 2.1. High-Fat Diet Significantly Increases the Body Weight of Mice

As shown in Figure 1A, there was no significant difference in body weight between the two groups at week 0. However, following 16 weeks of a high-fat diet (HFD16), mice in the HFD group exhibited a significant increase in body weight compared to mice in the ND group (*p* < 0.05), indicating that 16 weeks of a high-fat diet induced obesity in mice. When the high-fat diet was continued until 38 weeks (HFD38), the weight of the HFD mice was consistently higher than that of the ND group (Figure 1B), with body weight gains of 78.22% in the HFD group and 30.66% in the ND group. These results indicate that a long-term high-fat diet significantly increased mice body weight, whereas the long-term normal diet had little impact on mice body weight.

### 2.2. A High-Fat Diet Significantly Increases Serum Lipid Disorder and Liver Lipid Deposition in Mice

Compared with the ND group, the HFD significantly increased serum TC, LDL-C and HDL-C levels after 16 and 38 weeks (*p* < 0.05). However, it significantly reduced TG levels (Figure 2A–D). These results contrasted with those of Jeon et al. [15], which could be attributed to the differences in the high-fat diet composition of mice and individual animal variations, resulting in a slower increase in mouse serum triglyceride concentration. In this study, pathological changes in the mouse liver were also detected, and the results showed that the hepatocytes of the normal feed group were structurally intact, and no obvious fat vacuoles were observed. However, the high-fat diet group exhibited a large number of fat vacuoles in the hepatocytes, leading to serious fat accumulation. These changes were more severe in mice fed a HFD for 38 weeks than in mice fed a HFD for 16 weeks (Figure 2E), suggesting that 38 weeks of a high-fat diet caused lipid metabolism disorders in mice serum and fatty-like lesions in mouse liver, ultimately leading to NAFLD. These results indicated that a high-fat diet for 38 weeks successfully induced fat accumulation in the mice liver, causing lipoid lesions and leading to the occurrence and development of NAFLD.

### 2.3. Effect of a Long-Term High-Fat Diet on an Autophagy-Related Gene Expression Profile in NAFLD Mice

A total of 786 autophagy-related genes were included in this study. Based on this gene set, the autophagy expression profiles in the livers of mice fed a high-fat diet for 16 and 38 weeks were significantly different from those in the normal diet group. The PCA diagram (Figure 3A) showed that the expression levels of autophagy-related genes in the livers of mice in the high-fat group were different from those in the normal group, indicating that the high-fat diet significantly affected the systemic outline of autophagy genes in NAFLD mice.

Out of the 786 autophagy-related genes analyzed, 31 (3.9%) showed significant change after 38 weeks of high-fat diet induction, including 13 upregulated genes and 18 downregulated genes (Appendix A) (Figure 3B). Notably, among these 31 differentially expressed genes, *Txnip, lrs2, lgfbp2, Myc, Fgf21* and *Npy* were downregulated by 4-fold in the high-fat diet group, while *Pcsk9* and *Gadd45α* were upregulated by 4-fold, as shown in the heatmap of Figure 3C.

### 2.4. Functional Enrichment Analysis of GO and KEGG of Differentially Expressed Genes

To determine the biological function of the 31 identified DEGs, we performed GO and KEGG pathway enrichment analyses using R software. The results indicated that upregulated DEGs were mainly enriched in the molecular function (MF), biological process (BP) and cellular component (CC), while downregulated DEGs were mainly enriched in BP and MF. As shown in Figure 4A, upregulated DEGs were mainly associated with the negative regulation of cholesterol transport, negative regulation of sterol transport, endogenous apoptotic signaling pathway, positive regulation of neuronal apoptotic process, the apoptotic signaling pathway in endoplasmic reticulum stress, regulation of autophagy, process utilizing autophagy mechanism, autophagy and the positive regulation of autophagy. The upregulated DEGs were found in autophagosomes, COPII (Cytoplasmic coat protein complex II)-coated ER (Endoplasmic Reticulum) to Golgi transport vesicle, lysosome, lytic vacuole, endoplasmic reticulum protein-containing complex, endoplasmic reticulum quality control compartment, COPII vesicle coat, autolysosome, late endosome and coated vesicle. The downregulated DEGs included macroautophagy, chaperone-mediated autophagy, the positive regulation of autophagy, regulation of macroautophagy, regulation of anion transmembrane transport, and positive regulation of lipolysis. The TG metabolic process was related to the positive regulation of cellular catabolic processes. In the MF category, the downregulated DEGs were mainly associated with adenosine-nucleotidyl exchange factor activity, decanoate-CoA ligase activity, carbohydrate phosphatase activity, chaperone binding, unfolded protein binding, ubiquitin-like protein ligase binding, ubiquitin protein ligase binding, ATPase regulator activity and heat shock protein binding (Appendix A).

Enrichment analysis of the KEGG pathway revealed that downregulated differentially expressed genes were associated with the p53 signaling pathway, colorectal cancer, IL-17 signaling pathway, apoptosis, measles, cholesterol metabolism, Epstein–Barr virus infection, platinum drug resistance, small cell lung cancer and were related to toxoplasmosis. The upregulated differentially expressed genes were mainly enriched in the adipocytokine signaling pathway, regulation of lipolysis in adipocytes, and FoxO signaling pathway (Figure 4B).

### 2.5. PPI Network Visualization Results of Differential Genes and Hub Gene Screening

The PPI network of DEGs was constructed, consisting of 25 nodes (genes) and 36 edges (interactions) (Figure 5A). Using Cytoscape software and the analysis of seven algorithms including MCC, radio-activity, Stress, Dnnc, Degree, NNC, and BottleNeck, four genes were identified as hub genes: *Irs2*, *Pnpla2*, *Srebf2* and *Plin2* (Figure 5B). The expression levels of these four hub genes during high-fat diet feeding (16 weeks and 38 weeks) are shown in Figure 5C. The high-fat diet significantly reduced the expression of *Irs2* and *Plin2* at 16 and 38 weeks, but the differences in these two genes between the HFD16 and HFD38 groups were not significant (*p* > 0.05). Additionally, the expression of *Pnpla2* was significantly downregulated at 38 weeks of high-fat diet induction compared with the ND and HFD16-week mice. However, *Srebf2* were significantly upregulated after 38 weeks of high-fat diet induction and gradually increased with the progression of NAFLD.

### 2.6. Identification of Key Transcription Factor Modules

Transcription factors play a crucial role in regulating gene expression and function by binding to specific DNA sequences. Here, the iRegulon plug-in is used to predict the regulatory network of transcription factors and their target genes. Figure 6 shows the NES > 5 transcription factor modules. The *Tfdp1* transcription factor is predicted to regulate *Srebf2* and *Irs2*, while *Hoxa2* is predicted to regulate *Pnpla2* and *Irs2*.

### 2.7. Correlation Analysis between Hub Genes and Phenotype Changes

To investigate the association between phenotypic changes and the four hub genes, we analyzed the relationship between serum metabolic parameters, body weight and liver autophagy genes. The related heatmap shows that the four pivotal genes exhibit different degrees of correlation with serum metabolic indexes and genes under the induction of a high-fat diet. After being induced by a high-fat diet for 16 weeks, it was observed through correlation studies (Figure 7) that TC was positively correlated with LDL-C (R = 0.93). LDL-C is the main carrier for transporting endogenous cholesterol, and the correlation between TC and LDL-C was consistent with the results of blood lipid level determination in this study. In addition, TC was negatively correlated with *Plin2* and *Pnpla2* (R = −0.82, R = −0.7), which suggests that the accumulation of TC in mouse liver leads to a decrease in the expression of *Plin2* and *Pnpla2*. There is a significant positive correlation between TC and *Srebf2* (R = 0.76), indicating that as the severity of NAFLD increases, so does the expression of *Srebf2*. Additionally, the figure shows a negative correlation between the expression levels of *Srebf2* and both *Plin2* and *Pnpla2*.

### 2.8. Expression of Hub Gene in Pan-Cancer

To clarify the mRNA expression of four hub genes in pan-cancer, UALCAN combined with TCGA RNA-Seq data was used to analyze the differential expression patterns of *Irs2*, *Pnpla2*, *Srebf2* and *Plin2* in tumor and normal tissues. The results showed that the expression levels of these genes were significantly different in various tumor types compared to normal tissues. Specifically, *Irs2* was upregulated (*p* < 0.05) in colon adenocarcinoma (COAD), kidney chromophobe (KICH) and rectum adenocarcinoma (READ) but downregulated (*p* < 0.05) in bladder urothelial carcinoma (BLCA), breast invasive carcinoma (BRCA), cervical squamous cell carcinoma and endocervical adenocarcinoma (CESC), kidney renal papillary cell carcinoma (KIRP), lung adenocarcinoma (LUAD), lung squamous cell carcinoma (LUSC), pheochromocytoma and paraganglioma (PCPG) and uterine corpus endometrial carcinoma (UCEC). *Plin2* was significantly upregulated (*p* < 0.05) in glioblastoma multiforme (GBM), clear cell renal cell carcinoma (KIRC) and thyroid cancer (THCA). It was significantly downregulated (*p* < 0.05) in breast invasive carcinoma (BRCA), cholangiocarcinoma (CHOL), head and neck squamous cell carcinoma (HNSC), kidney chromophobe (KICH), liver hepatocellular carcinoma (LIHC), lung adenocarcinoma (LUAD), lung squamous cell carcinoma (LUSC), pheochromocytoma and paraganglioma (PCPG), prostate adenocarcinoma (PRAD) and uterine corpus endometrial carcinoma (UCEC). *Pnpla2* was significantly upregulated (*p* < 0.05) in liver hepatocellular carcinoma (LIHC), prostate adenocarcinoma (PRAD), thyroid carcinoma (THCA), and cholangiocarcinoma (CHOL), while significantly downregulated (*p* < 0.05) in breast invasive carcinoma (BRCA), head and neck squamous cell carcinoma (HNSC), kidney chromophobe (KICH), lung adenocarcinoma (LUAD), lung squamous cell carcinoma (LUSC) and pheochromocytoma and paraganglioma (PCPG). *Srebf2* was significantly upregulated (*p* < 0.05) in breast invasive carcinoma (BRCA), cholangiocarcinoma (CHOL), kidney chromophobe (KICH), liver hepatocellular carcinoma (LIHC), and lung squamous cell carcinoma (LUSC), while significantly downregulated (*p* < 0.05) in colon adenocarcinoma (COAD), glioblastoma multiforme (GBM), kidney renal clear cell carcinoma (KIRC), kidney renal papillary cell carcinoma (KIRP), lung adenocarcinoma (LUAD) and pheochromocytoma and paraganglioma (PCPG) (Figure 8).

## 3. Discussion

NAFLD is a chronic liver disease characterized by the excessive accumulation of lipids in hepatocytes and its exact mechanism is unknown. Although genetic factors have been shown to play a significant role in NAFLD development, the lack of validated genetic targets has hindered the development of effective and safe drugs to treat the disease. Therefore, biomarkers that are more sensitive and specific to NAFLD need to be further elucidated. Previous studies have shown that autophagy injury is closely related to NAFLD progression. Therefore, the objective of this study was to identify key autophagy differential genes and associated pathways that could serve as potential biomarkers or therapeutic targets for NAFLD.

To identify the expression of autophagy genes in NAFLD progression, we continued to feed the mice a high-fat diet for 38 weeks and observed a significant change in the expression profile of autophagy genes in the liver. We also screened 31 differentially expressed autophagy genes, of which 13 were upregulated and 18 were downregulated. Interestingly, among the 31 differentially expressed genes, *Txnip, lrs2, lgfbp2, Myc, Fgf21, Npy, Pcsk9* and *Gadd45α* were significantly regulated 4-fold by the high-fat diet. Previous studies have proven that these eight genes are closely related to the occurrence and development of NAFLD. *Fgf21* (fibroblast growth factor 21, an endocrine member in the fibroblast growth factor family) is mainly produced by the liver and adipose tissues and plays a key regulatory role in the body’s glycolipid metabolic balance. Increased levels of *Fgf21* have previously been found to be independent predictors of NAFLD in patients with NAFLD [16]. In addition, Singhal et al. also indicated more severe fibrosis and promoted NAFLD progression to HCC in mice lacking *Fgf21* after long-term exposure to an obese diet [17]. Furthermore, *Gadd45α*, a gene associated with lipogenesis, has been shown to be upregulated in liver cirrhosis, liver cancer, acute liver failure and NAFLD [18,19]. In this study, *Gadd45α* was increased 4-fold by the long-term high-fat diet, indicating that a long-term high-fat diet promotes fat synthesis and increases liver lipid accumulation. The data of Song et al. demonstrated that *Txnip* deficiency significantly inhibited starvation-induced autophagy in primary mouse hepatocytes and liver, indicating that *Txnip* might promote autophagy [3]. In this study, we found that *Txnip* was significantly downregulated after 38 weeks of a high-fat diet, which promoted lipid accumulation in hepatocytes, accelerated the susceptibility of mice to fatty hepatitis induced by a high-fat diet and aggravated NAFLD.

A total of 31 DEGs were selected in this study. To study the interaction between a series of genes and proteins to compensate for the fact that the expression of a single gene is not enough to explain the whole biological process and biological phenotypic changes, we performed enrichment analysis on 31 differentially expressed genes. Functional enrichment analysis indicated that the differentially expressed autophagy genes responsible for NAFLD were associated with multiple BPs, CCs, and MFs. For example, upregulated DEGs were mainly associated with the negative regulation of cholesterol transport. The apoptotic signaling pathway in ER stress was related to the response to ER stress, regulation of autophagy, and positive regulation of autophagy. The downregulated differentially expressed genes were mainly associated with carbohydrate phosphatase activity, ubiquitin-like protein ligase binding, ubiquitin protein ligase binding, ATPase regulator activity and heat shock protein binding. The DEG-associated KEGG signaling pathway analysis indicated that the upregulated genes were enriched in the p53 signaling pathway, colorectal cancer and the IL-17 signaling pathway. The downregulated DEGs were mainly enriched in the adipocytokine signaling pathway, regulation of lipolysis in adipocytes and FoxO signaling pathway. Regulatory signaling pathways for adipokines and lipolysis were downregulated compared with the ND group. This could explain why a chronic high-fat diet resulted in an imbalance in gene expression related to lipid metabolism, leading to an increase in TG content and serum cholesterol TC concentration. In this study, we found that the identified upregulated DEGs are associated with colorectal cancer, suggesting that these cancer-related DEGs may play an important role in the development of NAFLD-induced HCC. Research has shown that NAFLD is a risk factor for colorectal cancer. The incidence of colorectal tumors in NAFLD patients is twice as high as that in the general population [20]. NAFLD may be an independent risk factor for the occurrence of colorectal cancer, and the more severe the NAFLD lesions, the higher the risk of colorectal cancer. It is recommended that NAFLD patients, especially those with more severe lesions, undergo regular colonoscopy screening to detect colorectal tumors early and reduce mortality rates [21,22]. In addition, previous studies have shown that dysfunction of autophagy is an important component in the pathogenesis of NAFLD. In liver cells, autophagy can directly participate in lipid metabolism by degrading lipid droplets [23], and indirectly participate in lipid metabolism by maintaining the normal function of organelles and proteases involved in lipid metabolism [24]. There is also a close relationship between autophagy and inflammation. A high-fat diet can induce autophagy dysfunction in CD11c+ cells, increase the expression of pro-inflammatory cytokine IL-23 and promote the development of NAFLD [25]. The inhibition of the PI3K/Akt/mTOR signaling pathway can trigger autophagy, thereby reducing intracellular lipid accumulation and inflammation levels, and alleviating NAFLD [26]. Our research results, together with the literature reports, confirm that the activation of autophagy is negatively correlated with the regulation of lipid deposition and inflammatory responses.

Subsequently, a PPI network was constructed, and four hub genes, including *Srebf2, Pnpla2, Plin2* and *Irs2,* that were shown to be closely related to NAFLD were identified. In this study, the cholesterol synthesis gene *Srebf2* was significantly increased at 38 weeks with the prolongation of feeding on a high-fat diet. *Srebf2* is a key regulator of cellular cholesterol homeostasis which encodes sterol regulatory element binding protein 2 (SREBP2) [27]. Its activation promotes hepatocyte cholesterol accumulation by coordinately activating cholesterol biosynthesis and uptake and repressing cholesterol excretion [28]. *Srebf2* mRNA levels are significantly increased in patients with NASH, 3-fold higher than in non-NASH patients [29]. Cross-sectional human studies found an increased hepatic expression of SREBP2 and of its target genes, with the degree of SREBP2 activation paralleling the severity of hepatic cholesterol overload and liver histology [30]. These results indicated that a long-term high-fat diet promoted liver lipid accumulation and aggravated NAFLD by upregulating cholesterol synthesis genes. 

The accumulation of free fatty acids (FAs) is a critical trigger of lipid toxicity, leading to hepatocyte damage, inflammation and the progression from NAFLD to NASH. An increased flow of FAs from adipose tissue to the liver contributes to NAFLD development [31]. *Pnpla2* is a coding gene for lipocyte TG hydrolase (ATGL), also known as adipose triglyceride lipase, which catalyzes the initial step of triglyceride lipolysis in adipocytes, converting TGs into free fatty acids and leading to β-oxidation [32,33]. Zhou et al. found that the expression of *Pnpla2* in the liver of rats with NAFLD induced by a high-fat and high-fructose diet (HFFD) was significantly lower than that in rats fed a normal diet (*p* < 0.01) [34]. The study also showed a significant decrease in *Pnpla2* expression in the high-fat diet group compared with the normal diet group, with a positive correlation with the duration of high-fat diet feeding. These findings suggest that *Pnpla2* expression in the liver may decrease significantly with NAFLD development. Monitoring *Pnpla2* expression in the liver could therefore aid in the early diagnosis of NAFLD and in developing appropriate interventions to prevent disease progression.

Lipid droplet coating protein 2 (PLIN2) is the most abundant member of the PLIN protein family in both mouse and human steatotic livers and is highly expressed in most cells. Its expression is related to the severity of steatosis and the formation of fatty liver in NAFLD [35,36]. An increased *Plin2* expression is generally associated with increased intracellular lipid droplet formation, growth and fusion [37]. Our study found that the expression of *Plin2* in the mice liver was downregulated after 38 weeks of induction by a high-fat diet, which was speculated to be related to liver-initiated lipid-phagocytic feedback regulation under a high-fat diet [38]. The specific molecular mechanism by which *Plin2* regulates hepatic lipid accumulation under physiological and pathophysiological conditions remains unclear. Therefore, this study mainly revealed the correlation between high-fat diet induction and the downregulation of mouse liver *Plin2*, and more studies are needed to reveal the regulatory effects of *Plin2* and autophagy in NAFLD development.

In this study, the *Irs2* in the liver of the model group was significantly downregulated 4-fold under the influence of a 38-week long-term high-fat diet, consistent with previous reports of *Irs2* downregulation in obese individuals and patients with NAFLD [39,40]. Studies have confirmed that there is a correlation between IR and autophagy. When autophagy is inhibited, it leads to a large amount of accumulation of damaged organelles, such as mitochondria, and the generation of accumulation of metabolites, such as reactive oxygen species, which in turn affects the conduction of insulin signals and leads to IR [41]. Meanwhile, the results of this study also showed that although the gene expression of *Irs2* in mice with mild NAFLD at 16 weeks and severe NAFLD at 38 weeks was significantly lower than that in the ND group, there was no significant difference in the gene expression of *Irs2* in the HFD16 and HFD38 weeks (*p* > 0.05). The above results are very similar to those obtained by Honma et al. [42]. Taken together, these findings suggested that changes in insulin signaling molecules were associated with hepatic changes in simple steatosis rather than NAFLD progression.

In addition, we found that two transcription factors, *Tfdp1* and *Hoxa2*, may be related to the progression of NAFLD into HCC. The TFDP family comprises dimeric ligands for transcription factors. The TFDP family and E2F family together play important functions in regulating the cell cycle, cell differentiation and apoptosis, with *Tfdp1* playing a crucial role in mediating diseases by affecting the CDK-RB-E2F cell cycle regulation axis. Previous studies have shown significant increases in E2F1 expression in the liver of obese humans and fatty liver mice. In the db/db mouse model of NAFLD, the deletion of E2f1 prevents lipid accumulation in the liver, which may be related to the improvement in hepatic metabolic homeostasis by E2F1 through the control of glucose and lipid production pathways [43,44]. Homeobox genes (HOX) play an important role in biological development. *Hoxa2* is part of the HOXA transcription factor family. The protein encoded by the *Hoxa2* possesses both DNA binding and transcription factor activities. It plays a crucial role in regulating the proliferation and differentiation of embryonic and adult cells by binding to specific sites on DNA and initiating the transcription of target genes. Studies have shown that the expression of the *Hoxa2* in liver cancer samples is significantly upregulated [45], and in the progression of mild fibrosis and severe fibrosis, *Hoxa2* DNA methylation is significantly increased [46]. These results suggested that *Hoxa2* might play a role in promoting the progression of hepatocellular carcinoma and become a potential target for the treatment of hepatocellular carcinoma. 

The results of pan-cancer analysis in this study describe the expression levels of four genes *(Irs2*, *Plin2*, *Pnpla2* and *Srebf2*) in various types of cancerous tissues compared to normal tissues. The statistical analysis revealed that the expression levels of these genes were significantly different in different types of cancer. These findings could be used to further explore the role of these genes in the development and progression of cancer. The upregulation of a gene in a particular cancer type could suggest that the gene is involved in the development or progression of that cancer, while the downregulation could suggest that the gene may act as a tumor suppressor in that cancer [47]. However, further experiments are needed to confirm the role of these genes in cancer.

## 4. Materials and Methods

### 4.1. Animals

Male specific-pathogen-free (SPF) grade C57BL/6J mice, aged 42–56 days and weighing 23–25 g, were obtained from Beijing Huafukang Biotechnology Co., Ltd. with license number SCXK (Beijing, China) 2014-0004, and passed the examination by the China Medical Laboratory Animal Research Institute. The mice were adaptively fed for 7 days with free access to food and water under an SPF grade environment. The day and night were balanced with a 12-h period, and the temperature and humidity were maintained at 22 °C ± 1 °C and 55% ± 5%, respectively. The mice were then divided into the normal diet (ND) and high-fat diet (HFD) groups (*n* = 10 per group). The high-fat diet (53% kcal from fat) was formulated based on previous publications [48]; its composition is displayed in Appendix A. Body weight and food intake were measured weekly during the experiment, and the mice were fed for 16 and 38 weeks to represent the mild and severe NAFLD mouse model [13]. At the end of the experiment, the mice were sacrificed, and serum and liver samples were collected. Animal experiments were conducted in full compliance with animal ethical requirements and approved by the Animal Ethics Review Committee of Zunyi Medical University (ZMU21-2203-433).

### 4.2. Biochemical Determination

Serum total cholesterol (TC), triglyceride (TG), low-density lipoprotein cholesterol (LDL-C), and high-density lipoprotein cholesterol (HDL-C) levels were determined using commercial kits purchased from Nanjing Jiancheng Biotechnology Co., Ltd (Nanjing, China). and tested by an ELISA reader.

### 4.3. Histopathological Examination of the Liver (HE Staining)

The fixed liver tissues were dehydrated with a series of alcohol concentrations from low to high (70%, 80%, 95%, 100%), cleared in xylene, embedded in wax, cut into 5 μm thick sections, and dried at 65 °C. After equilibration for 10 min, the wax was dissolved in xylene, rehydrated in different concentrations of alcohol from high to low (100%, 95%, 90%, 80% and 70%), and finally hydrated in distilled water. The sample was then stained with a safranin solution for 4 min, rinsed in running water for 10 s, differentiated in a solution of hydrochloric acid and ethanol for 3 s, followed by rinsing in running water until the blue color disappeared. Finally, the sample was stained with eosin solution for 2 min and rinsed in water for an additional 2 min. The stained sections were then successively dehydrated in 80% and 90% ethanol for several seconds and in anhydrous ethanol for 1 min, followed by sealing and microscopic examination.

### 4.4. Total RNA Extraction and Sequencing

To extract total RNA, 20 mg of liver tissue stored at −80 °C was weighed and added to 1 mL of Trizol reagent. The sample was homogenized on ice and incubated at room temperature for 5 min. Then, 200 μL of chloroform was added and the mixture was incubated at room temperature for 10 min. After centrifugation at 12,000× *g* for 15 min, the RNA was washed with 75% alcohol, centrifuged at 7500× *g* for 15 min and dissolved in 30 μL of DEPC water. The RNA concentration and quality were determined using an ultramicro spectrophotometer. RNA sequencing was performed using the BGI Seq-500 sequencing platform of the Beijing Genomics Institute (BGI) of China National Gene Corporation.

### 4.5. Screening of Autophagy-Related Genes

Autophagy-related genes were selected through the Online Mendelian Inheritance in Man (OMIM), the National Center for Biotechnology Information (NCBI) database and the Human Autophagy Database (HADb). The search keywords were “mouse”, “Autophagy” and “gene”. The screened autophagy-related genes were compared in the NCBI literature, and a total of 786 autophagy-related genes were screened. The expression data for these genes were extracted from the expression matrix and analyzed.

### 4.6. Analysis of Transcriptome Gene Expression

The extracted 786 autophagy-related gene expression matrix data were further analyzed using R language. After filtering out low-expression genes based on the CPM values, the gene expression data set was subjected to TMM standard normalization processing using the calcNormFactors function in the edgeR package. After filtering, a negative binomial log-linear model statistical test was performed to screen out the differentially expressed genes in the 38-week high-fat diet group and the control group. Data visualization was conducted through a volcano diagram. The screening criteria for differential genes were FDR < 0.05, |log2(Fold Change)|≥ 1.

### 4.7. Functional Enrichment of Differentially Expressed Genes (DEGs)

The enrichment of GO function and the KEGG pathway of DEGs was analyzed by using R language ggplot2. The R package “org.Mm.e.g.,db” was used to convert the DEG symbol “SYMBOL” to “ENTREZID”, and the R software packages “clusterProfiler” and “ggplot2” were used for GO functional enrichment and the KEGG pathway enrichment analysis of DEGs. The GO database annotates gene products from three aspects of biology: molecular function (MF), biological process (BP), and cellular component (CC). KEGG was used to analyze the potential function of DEGs in the signaling pathway.

### 4.8. Analysis of Protein–Protein Interaction (PPI)

The STRING database (www.string-db.org accessed on 14 February 2023) was used to construct a PPI network of DEGs with a minimum interaction score of 0.4. The network atlas was drawn by Cytoscape software (Version 3.9.0) under the JAVA program. In addition, the plug-in Cytoscape, cytoHubba, was used to select the top 10 genes according to seven algorithms, MCC, Radiality, Stress, Dnnc, Degree, NNC and BottleNeck. Then, a Venn diagram was drawn with the R software package “Venn diagram” to obtain the intersecting genes, which were regarded as hub genes.

### 4.9. Transcription Factors Analysis

The regulation of gene expression was mediated by transcription factors. To identify and visualize key transcription factors for the core genes in the PPI networks, we used the iRegulon plug-in of Cytoscape. Transcription factors with a normalized enrichment score (NES) greater than 5 were selected.

### 4.10. Data Processing

According to the needs of the experimental data, R software (Version 4.1.0) was used to analyze the significant differences between the two groups with a *t*-test, and one-way analysis of variance was used to compare the differences between multiple groups. The R software was also used to generate graphs, while Cytoscape software (Version 3.9.0) was used to construct the network. A *p*-value or adjusted *p*-value less than 0.05 was considered statistically significant.

## 5. Conclusions

In summary, we conducted an analysis of autophagy gene expression profiles in the liver of NAFLD mice after 16 and 38 weeks of a high-fat diet and found that a variety of physiological activities related to NAFLD, especially lipid and glucose metabolism, were affected by the high-fat diet. In addition, following the analysis of the GO and KEGG pathways, four genes involved in glucose and lipid metabolism—*Srebf2*, *Pnpla2*, *Plin2* and *Irs2*—were identified by the PPT network. Upregulating the cholesterol synthesis gene *Srebf2* may exacerbate NAFLD by promoting hepatic lipid accumulation. Conversely, downregulating the fatty acid oxidation genes *Pnpla2* and *Plin2* may promote the formation of intracellular lipid droplets, while downregulating the *Irs2* gene may induce insulin resistance and inhibit autophagy in liver cells, ultimately triggering NAFLD. These four DEGs and related pathways could be used as potential diagnostic and therapeutic targets for NAFLD. However, further studies are necessary to elucidate the regulatory effects of these four genes in NAFLD. Additionally, our study has some limitations, such as the fact that the identified DEGs and related pathways were generated only by bioinformatics analysis. Thus, these findings need to be further confirmed in future studies in NAFLD mice and clinical specimens. Despite these limitations, our results provide a basis for further evaluating the key autophagy genes and related pathway-mediated mechanisms in NAFLD.

## Figures and Tables

**Figure 1 ijms-24-06437-f001:**
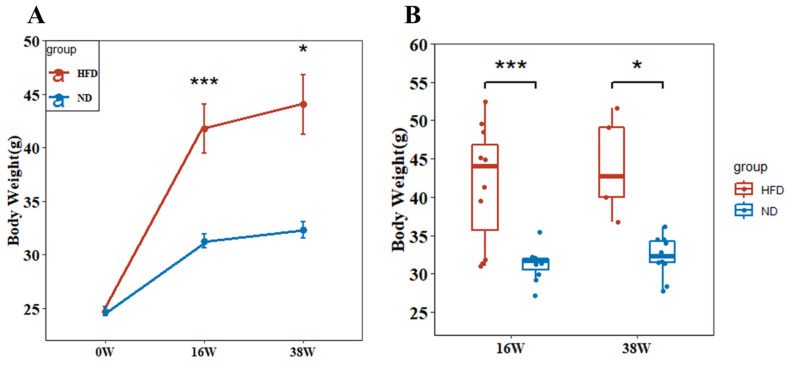
Mouse body weight. (**A**) Body weight curves of mice in each group were recorded during the experiment. (**B**) The HFD group exhibited a significantly higher body weight gain after 38 weeks of a high-fat diet compared to the ND group. (* *p* < 0.05, *** *p* < 0.001).

**Figure 2 ijms-24-06437-f002:**
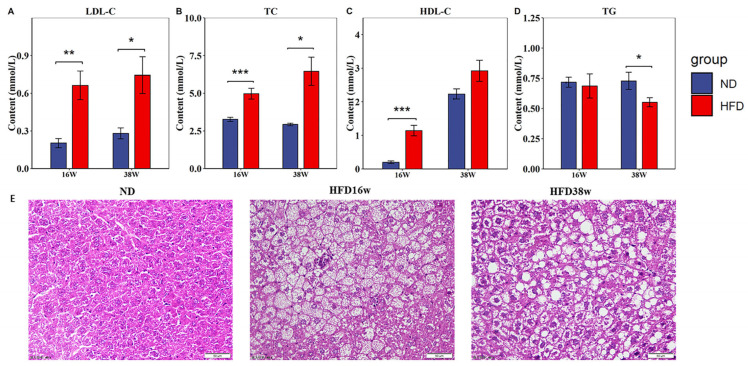
Effects of a high-fat diet on lipid metabolism in mice. (**A**–**D**) Thirty-eight weeks of a high-fat diet significantly increased the levels of serum LDL-C, TC, HDL-C A-D and TG in the model group. (**E**) HE staining of liver tissues in the model group and the control group showed that there were fat vacuoles in the liver, which caused liver lipid accumulation (400×) (* *p* < 0.05, ** *p* < 0.01, *** *p* < 0.001).

**Figure 3 ijms-24-06437-f003:**
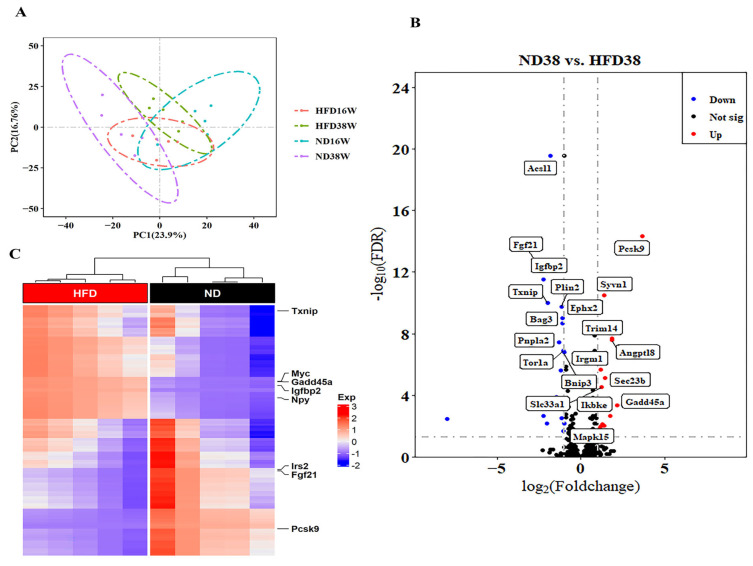
Expression profiles of 786 autophagy genes in the liver after 38 weeks of a high-fat diet. (**A**) Principal component analysis score chart; (**B**,**C**) Volcanic and heatmaps of identified differentially expressed genes (DEGs) between the livers of mice fed a high-fat diet for 38 weeks and the livers of mice fed a normal diet. The volcanic diagram showed a total of 31 differentially expressed genes, with red dots representing 13 genes upregulated based on the adjusted *p* value FDR < 0.05 log2 (fold change) > 1. The blue dots represent 1 of the 18 downregulated genes for FDR < 0.05 and log2 (fold change) < 1. Black dots indicate genes whose expression is not significantly different. The boxes in the figure identify the upregulated and downregulated genes in the top ten in FDR order.

**Figure 4 ijms-24-06437-f004:**
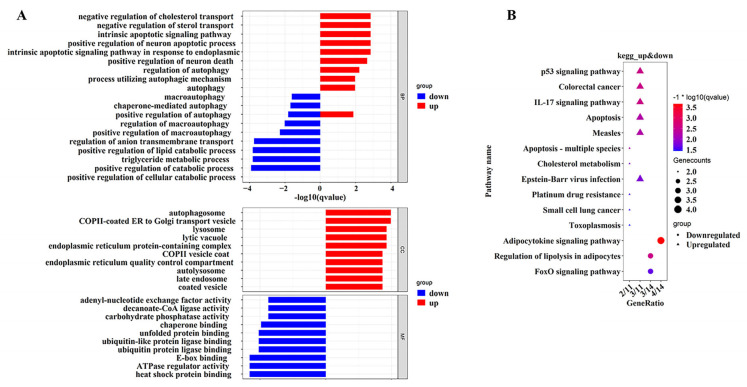
Functional enrichment analysis. (**A**) GO enrichment analyses of 31 common DEGs. The top 10 terms in each GO category (MF, molecular function; CC, cellular components; BP, biological processes). Red represents up-regulated genes and blue represents down-regulated genes; (**B**) KEGG enrichment analyses of 31 common DEGs. All significant KEGG pathways. The cut-off standard is FDR < 0.05. The color represents the FDR of this enrichment pathway, and the red represents a more prominent FDR. Gene ratio refers to the proportion of genes enriched in each pathway. Triangles represent upregulated genes, and dots represent downregulated genes.

**Figure 5 ijms-24-06437-f005:**
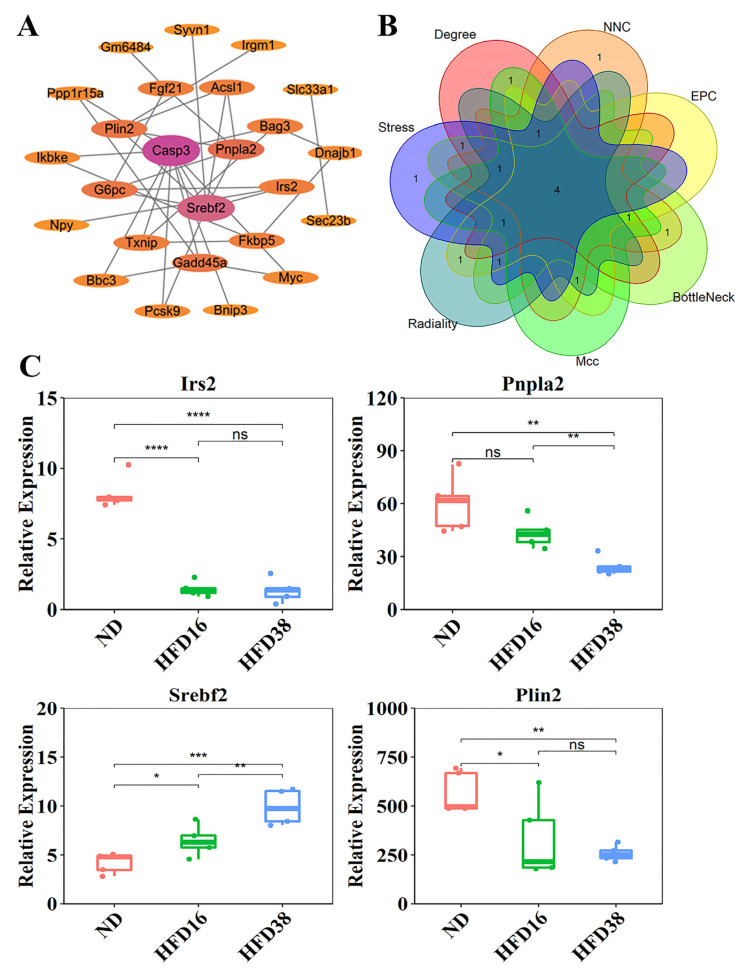
Screening of protein interactions (PPIs) and hub genes. (**A**) PPI network of 31 DEGs. The minimum required interaction score is 0.4, consisting of 25 nodes (genes) and 36 edges (interactions); (**B**) screening four hub genes based on seven algorithms; (**C**) hub gene *Irs2, Pnpla2, Srebf2*, and *Plin2* expression. The expression of *Irs2, Pnpla2* and *Plin2* was significantly downregulated in the 38-week high-fat diet, while the expression of *Srebf2* was significantly upregulated in the 38-week high-fat diet. * *p* < 0.05; ** *p* < 0.01; *** *p* < 0.001; **** *p* < 0.0001; ns *p* > 0.05.

**Figure 6 ijms-24-06437-f006:**
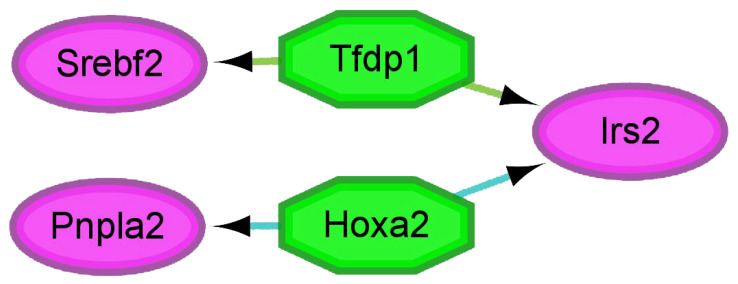
Transcription factor target network for NES > 5 using the iRegulon plug-in. The green octagonal node represents the predicted transcription factor. Pink oval nodes represent genes regulated by transcription factors.

**Figure 7 ijms-24-06437-f007:**
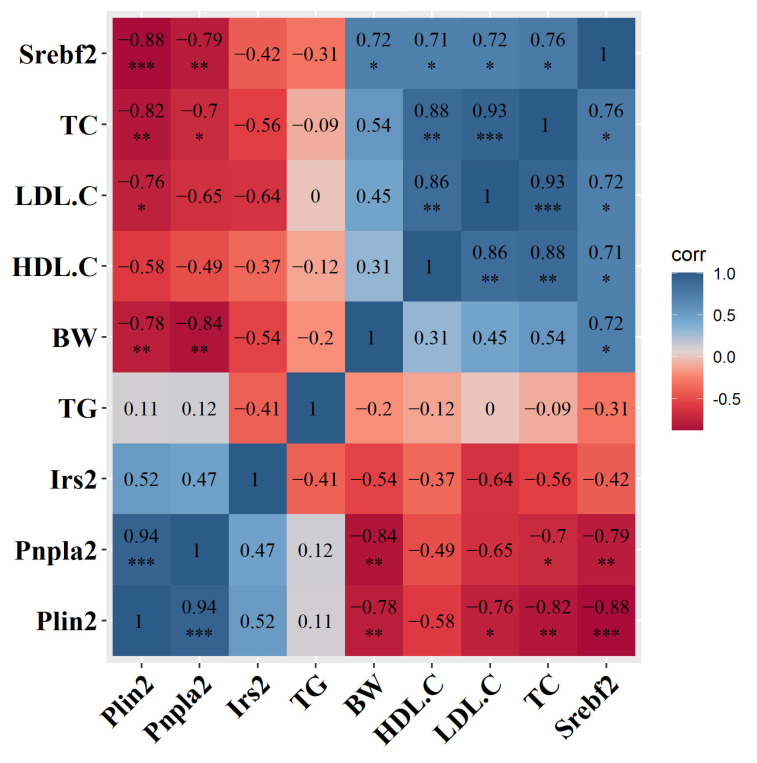
Contour visualization of the interaction between high-fat diet and liver autophagy gene expression and the correlation between clinical indicators. * *p* < 0.05; ** *p* < 0.01; *** *p* < 0.001.

**Figure 8 ijms-24-06437-f008:**
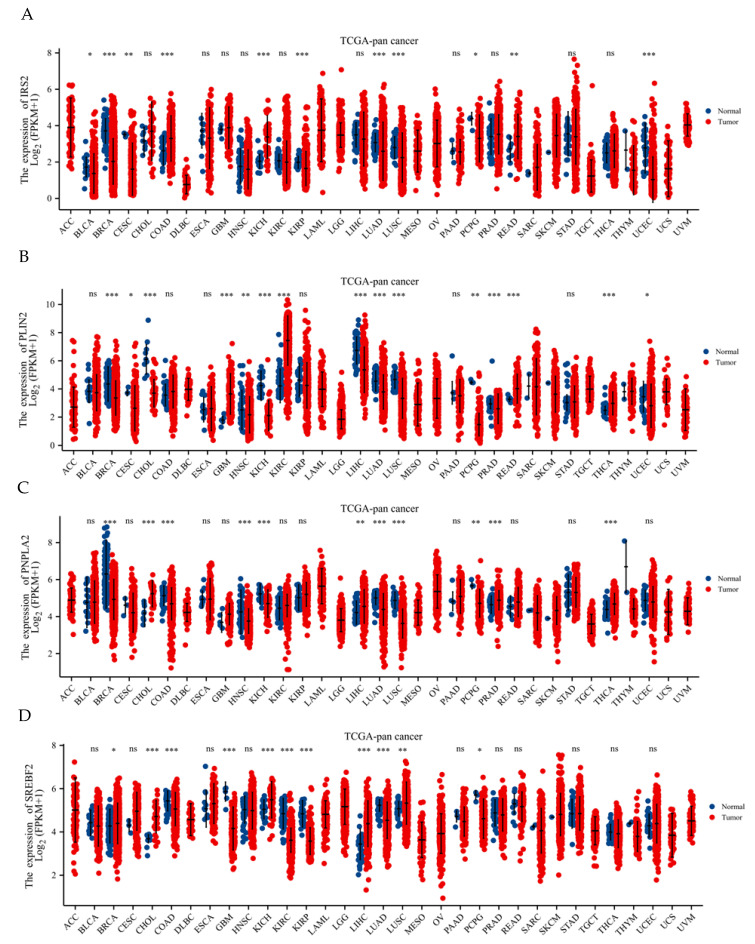
Pan-cancer analysis of the expression of four hub genes. (**A**) Pan-cancer analysis of *Irs2*; (**B**) pan-cancer analysis of *Pnpla2*; (**C**) pan-cancer analysis of *Plin2Srebf2*. (**D**) Pan-cancer analysis of *Srebf2*. * *p* < 0.05; ** *p* < 0.01; *** *p* < 0.001; ns *p* > 0.05.

## Data Availability

The authors confirm that the data supporting the findings and conclusions of this study are available in the article and its Appendix A.

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
