# Peer review of "Transcriptome Sequencing Reveals Autophagy Networks in Rat Livers during the Development of NAFLD and Identifies Autophagy Hub Genes"

_ijms, 2023, doi:10.3390/ijms24076437_

Round 1
Reviewer 1 Report
Manuscript ID: ijms-2254534
Title: Transcriptome sequencing reveals autophagy networks in rat livers during the development of NAFLD and identifies autophagy hub genes.
Authors: Jian Xie, Qiuyi Chen,Yongxia Zhao, Mingxia Luo, Xin Zen, Lin Qin, Daopeng Tan and Yuqi He
First of all, I want to mention that the manuscript did not have numbered lines and from this point of view it was difficult to revise.
The manuscript entitled “Transcriptome sequencing reveals autophagy networks in rat livers during the development of NAFLD and identifies autophagy hub genes” reports the biological function of the autophagy hub gene, which the authors argue that could be used as a potential therapeutic target and diagnostic marker of nonalcoholic fatty liver disease. The authors reported 31 differentially expressed autophagy-related genes, of which 13 were upregulated and 18 downregulated. The work is of importance since the results provide a basis for further evaluation of key autophagy genes and related pathway-mediated mechanisms in nonalcoholic fatty liver disease.
I recommend the manuscript to be considered for publication after minor revision.
My concerns are:
General: The abbreviated phrases should be written in full the first time they are used. So, please specify full name for all phrases when they are first time used, e.g: SPF, COPII, PPI, etc. Moreover, the authors use both the abbreviation and the full description for the same phrase in several parts of the manuscript, e.g: fatty acids (FAs). Please correct this aspect.
Authors may use italics for mathematical symbols, e.g: p values. In some parts of the manuscript, the p value is in italics and in other parts, it is not italic, I ask the authors to correct this aspect.
The italics should be used for gene names. In some parts of the manuscript, gene names is in italics and in other parts, it is not italic, I ask the authors to correct this aspect.
The use of English language should be improved throughout the manuscript.
Abstract
Please revise the third sentence from (2) Methods. Probably the authors wanted to say: The potential biological and pathological functions of DEGs were investigated by GO and KEGG analyses.
Any abbreviations used in the abstract must be defined also here.
Introduction
The sentence: However, this method has a higher accuracy and repeatability in the diagnosis of NAFLD, and it also has larger economic use costs and complications. Please specify which method the authors are referring to.
In the second paragraph of introduction chapter, the italics should be used for gene names.
Materials and Methods
2.3. Histopathological examination of the liver (HE staining)
The penultimate sentence must be revised
2.4. Total RNA extraction and sequencing
The first three sentences must be revised
Results
For the presentation of the results, please do not mix the tenses.
3.1. High-fat diet significantly increases the body weight of mice
The first sentence must be revised.
Please describe the results obtained and shown by figure 8. No results are presented regarding figure 8.
Discussion
For the third paragraph from the discussion chapter, I ask the authors to compare their results with those obtained in the literature; no bibliographic reference is specified here.
The last paragraph refers to figure 8. Please include this results in the results chapter.
Author Response
Dear professor:
Thank you very much for your kindly comments on our manuscript titled "Transcriptome sequencing reveals autophagy networks in rat livers during the development of NAFLD and identifies autophagy hub genes ". We appreciate your time and effort in reviewing our work. Based on your suggestions, we carefully revised the manuscript. We are now sending the revised article for your re-consideration to publish in “Intemational Joural of Molecular Sciences”. Below you will find our point-by-point responses to the reviewers’ comments and questions, the original comments are in black, and our responses are in red. We believe that your comments have helped us to improve our manuscript and look forward to hearing from you soon for a favorable decision.
Thank you again for your time and consideration.
Yours sincerely,
Jian Xie
1. The abbreviated phrases should be written in full the first time they are used. So, please specify full name for all phrases when they are first time used, e.g: SPF, COPII, PPI, etc. Moreover, the authors use both the abbreviation and the full description for the same phrase in several parts of the manuscript, e.g: fatty acids (FAs). Please correct this aspect.
Answer: We sincerely appreciate the valuable comments. We are very sorry for the mistakes in the manuscript and inconvenience they caused in your reading. We have modified the abbreviations throughout the text to ensure that the full name is used upon first mention.
2. Authors may use italics for mathematical symbols, e.g: p values. In some parts of the manuscript, the p value is in italics and in other parts, it is not italic, I ask the authors to correct this aspect.
Answer: We sincerely appreciate the valuable comments. We have already used italic font for all p-values in the manuscript.
3. The italics should be used for gene names. In some parts of the manuscript, gene names is in italics and in other parts, it is not italic, I ask the authors to correct this aspect.
Answer: Thank you for your review and feedback on the manuscript. Regarding the use of italics for gene names, we will make the necessary changes based on your suggestion. We will ensure consistency in the formatting of gene names throughout the manuscript by making all gene names italicized or non-italicized as appropriate. Thank you again for your guidance.
4. The use of English language should be improved throughout the manuscript.
Answer: Thank you for your feedback and suggestions. We have further improved and enhance the English language expression in the manuscript during the revision process, carefully checked the accuracy of grammar, spelling, and words, and ensured that the language expression of the entire manuscript is more accurate, clear, and standardized, so as to better convey our research results.
5. Abstract: Please revise the third sentence from (2) Methods. Probably the authors wanted to say: The potential biological and pathological functions of DEGs were investigated by GO and KEGG analyses. Any abbreviations used in the abstract must be defined also here.
Answer: Thank you very much for your critical comments regarding this issue. We have revised this sentence to “Gene ontology (GO) and Kyoto Encyclopedia of Genes and Genomes (KEGG) analysis were performed to investigate the potential biological and pathological functions of DEGs.”
6. Introduction. The sentence: However, this method has a higher accuracy and repeatability in the diagnosis of NAFLD, and it also has larger economic use costs and complications. Please specify which method the authors are referring to.
Answer: Thank you for the comments. This “method” refers to “non-invasive imaging diagnosis method”. And we have revised this sentence to “In recent years, noninvasive imaging diagnosis methods have been developed, while this method has a greater accuracy and repeatability in the diagnosis of NAFLD, and it also come with higher economic costs and a risk of complications”. So as to avoid readers' ambiguity.
7. In the second paragraph of introduction chapter, the italics should be used for gene names.
Answer: We appreciate your feedback and have made the necessary revisions to the manuscript. The gene names have been italicized in the manuscript.
8. Materials and Methods. 2.3. Histopathological examination of the liver (HE staining) ,The penultimate sentence must be revised.
Answer: Thank you for the comments. I'm sorry that the reading experience is not good. We have revised the sentence to “The sample was then stained with a safranin solution for 4 minutes, rinsed in running water for 10 seconds, differentiated in a solution of hydrochloric acid and ethanol for 3 seconds, followed by rinsing in running water until the blue color disappeared. Finally, the sample was stained with eosin solution for 2 minutes and rinsed in water for an additional 2 minutes.”
9. 2.4. Total RNA extraction and sequencing. The first three sentences must be revised.
Answer: Thank you for the comments. I'm sorry that the reading experience is not good. We have revised the sentence to “To extract total RNA, 20 mg of liver tissue stored at -80°C was weighed and added to 1 mL of Trizol reagent. The sample was homogenized on ice and incubated at room temperature for 5 minutes. Then, 200 μL of chloroform was added and the mixture was incubated at room temperature for 10 minutes. After centrifugation at 12000g for 15 minutes, the RNA was washed with 75% alcohol, centrifuged at 7500 × g for 15 minutes, and dissolved in 30 μL of DEPC water.”
10. Results. For the presentation of the results, please do not mix the tenses.
Answer: Thank you for your feedback. The tenses in the article have been modified to ensure that the tenses we use when presenting the results are consistent and appropriate.
11. 3.1. High-fat diet significantly increases the body weight of mice. The first sentence must be revised.
Answer: Thank you for the comments. We have revised the sentence to “As shown in Figure 1A, there was no significant difference in body weight between the two groups at 0 week. However, following 16 weeks of high-fat diet feeding (HFD16), mice in the HFD group exhibited a significant increase in body weight compared to mice in the ND group (p < 0.05), indicating that 16 weeks of high-fat diet induced obesity in mice.”
12. Please describe the results obtained and shown by figure 8. No results are presented regarding figure 8.
Answer: Thank you for your critical comments regarding this issue. The result of Figure 8 has been added to the article, and the following is the added content. “To clarify the mRNA expression of four hub genes in pan-cancer, UALCAN combined with TCGA RNA-Seq data was used to analyze the differential expression patterns of Irs2, Pnpla2, Srebf2, and Plin2 in tumor and normal tissues. The results showed that the expression levels of these genes were significantly different in various tumor types compared to normal tissues. Specifically, Irs2 was upregulated (p < 0.05) in colon adenocarcinoma (COAD), kidney chromophobe (KICH), and rectum adenocarcinoma (READ) but downregulated (p < 0.05) in bladder urothelial carcinoma (BLCA), breast invasive carcinoma (BRCA), cervical squamous cell carcinoma and endocervical adenocarcinoma (CESC), kidney renal papillary cell carcinoma (KIRP), lung adenocarcinoma (LUAD), lung squamous cell carcinoma (LUSC), pheochromocytoma and paraganglioma (PCPG), and uterine corpus endometrial carcinoma (UCEC). Plin2 is significantly upregulated (p < 0.05) in glioblastoma multiforme (GBM), clear cell renal cell carcinoma (KIRC), and thyroid cancer (THCA). It is significantly downregulated (p < 0.05) in breast invasive carcinoma (BRCA), cholangiocarcinoma (CHOL), head and neck squamous cell carcinoma (HNSC), kidney chromophobe (KICH), liver hepatocellular carcinoma (LIHC), lung adenocarcinoma (LUAD), lung squamous cell carcinoma (LUSC), pheochromocytoma and paraganglioma (PCPG), prostate adenocarcinoma (PRAD), and uterine corpus endometrial carcinoma (UCEC). Pnpla2 was significantly upregulated (p < 0.05) in liver hepatocellular carcinoma (LIHC), prostate adenocarcinoma (PRAD), thyroid carcinoma (THCA), and cholangiocarcinoma (CHOL), while significantly downregulated (p < 0.05) in breast invasive carcinoma (BRCA), head and neck squamous cell carcinoma (HNSC), kidney chromophobe (KICH), lung adenocarcinoma (LUAD), lung squamous cell carcinoma (LUSC), and pheochromocytoma and paraganglioma (PCPG). Srebf2 was significantly upregulated (p < 0.05) in breast invasive carcinoma (BRCA), cholangiocarcinoma (CHOL), kidney chromophobe (KICH), liver hepatocellular carcinoma (LIHC), and lung squamous cell carcinoma (LUSC), while significantly downregulated (p < 0.05) in colon adenocarcinoma (COAD), glioblastoma multiforme (GBM), kidney renal clear cell carcinoma (KIRC), kidney renal papillary cell carcinoma (KIRP), lung adenocarcinoma (LUAD), pheochromocytoma and paraganglioma (PCPG).”
13. Discussion. For the third paragraph from the discussion chapter, I ask the authors to compare their results with those obtained in the literature; no bibliographic reference is specified here.
Answer: Thank you for your critical comments regarding this issue. The results have been compared with references in the manuscript, and the modifications are as follows.:“previous studies have shown that dysfunction of autophagy is an important component in the pathogenesis of NAFLD. In liver cells, autophagy can directly participate in lipid metabolism by degrading lipid droplets [24], and indirectly participate in lipid metabolism by maintaining the normal function of organelles and proteases involved in lipid metabolism [25]. There is also a close relationship between autophagy and inflammation. A high-fat diet can induce autophagy dysfunction in CD11c+ cells, increase the expression of pro-inflammatory cytokine IL-23, and promote the development of NAFLD [26]. Inhibition of the PI3K/Akt/mTOR signaling pathway can trigger autophagy, thereby reducing intracellular lipid accumulation and inflammation levels, and alleviating NAFLD [27]. Our research results, together with literature reports, confirm that the activation of autophagy is negatively correlated with the regulation of lipid deposition and inflammatory responses.”
- Singh, R.; Kaushik, S.; Wang, Y.; Xiang, Y.; Novak, I.; Komatsu, M.; Tanaka, K.; Cuervo, A.M.; Czaja, M.J. Autophagy regulates lipid metabolism. Nature 2009, 458, 1131-1135, doi: 10.1038/nature07976.
- Byrnes, K.; Blessinger, S.; Bailey, N.T.; Scaife, R.; Liu, G.; Khambu, B. Therapeutic regulation of autophagy in hepatic metabolism. Acta Pharm Sin B 2022, 12, 33-49, doi: 10.1016/j.apsb.2021.07.021.
- Galle-Treger, L.; Helou, D.G.; Quach, C.; Howard, E.; Hurrell, B.P.; Muench, G.; Shafiei-Jahani, P.; Painter, J.D.; Iorga, A.; Dara, L.; et al. Autophagy impairment in liver CD11c(+) cells promotes non-alcoholic fatty liver disease through production of IL-23. Nat. Commun. 2022, 13, 1440, doi: 10.1038/s41467-022-29174-y.
- Sun, C.; Zhang, J.; Hou, J.; Hui, M.; Qi, H.; Lei, T.; Zhang, X.; Zhao, L.; Du H. Induction of autophagy via the PI3K/Akt/mTOR signaling pathway by Pueraria flavonoids improves non-alcoholic fatty liver disease in obese mice. Biomed. Pharmacother. 2023, 157, 114005, doi: 10.1016/j.biopha.2022.114005.
14. The last paragraph refers to figure 8. Please include this results in the results chapter.
Answer: Thank you for the comments. The content of this part has been put into the results 3.8.
Reviewer 2 Report
Abstract- Please the authors must rewrite abbreviations in full names for example ND38, HFD 38, PPI must be put inside brackets. Too many abbreviations make hard to understand the meaning for readers not so knowledgeable on the subject. If there are too many words i suggest inserting a generic term and describe details in the text (Materials and Methods). Genes firstly indicated here, must be written in full than abbreviated in brackets.
1 Introduction-Page 2-
Please delete “aging” the sentence must sound as “Autophagy can degrade dead organelles…” and insert proper References here.
Again, please rewrite in full genes indicated as Sox9, Ccl20, Cxcl1, Cd24---and also Nos3, Igf1, Vamp8, Fos, Hmox1.
GSE89632 abbreviation what means?
Please check the order of the words. The sentence at the end of Introduction must be rewritten as “ Therefore, we constructed a high-fat diet-fed mouse model for short-term (16 weeks) or long-term ( 38 weeks), and induced NAFLD.”
2 Materials and Methods
-Page 2-Please indicate the provider of different diets adopted in the study and the formulation of fat (coconut oil, soybean oil, lard) or a Reference indicating the information, which can be useful for other researchers interested in NAFLD. Different diet composition in fat even if provide similar calories might exert different effects in hepatic steatosis.
Page 4-Indicate in full “TF analysis”
I suggest to estimate by image analysis program lipid droplets number inside H&E stained liver sections. This is important to reinforce microscopic feature of steatosis shown in Figure 2.
3 Results
Page 8 Figure 4 must be enlarged because it is hard to understand . Please insert a brief explanation in Figure 4 legend.
Page 9 Please insert full names for Tfdp1, Srebf2, Irs2, Hoxa2,Pnpla2,Irs2. These abbreviations must be put in the list of abbreviations at the end of the text.
Page 11 Please insert Figure 8 with more indications in Figure legend as Supplementary file (S1). There are too many data in the text and they are confounding.
4 Discussion
Page 11-Please delete some sentences that have been just indicated in the Introduction and redundant. For example “ “At present, the diagnosis of NAFLD mainly depends on biochemical indicators, imaging and histopathological examination”.
Please the authors might indicate which type of autophagy has been defined by genic profile, ie non selective macro autophagy, or chaperone-mediated autophagy? Both mechanisms are deeply involved in the liver.
Page 12 the relationship between colorectal cancer and NAFLD must be better discussed with appropriate References.
Page 13-Similarly, the late increase of Srebf2 at 38 weeks of HFD must be better discussed here with relative References.
Page 14- the English txt is understandable. Please rewrite the sentence “ The gene expression protein has DNA bi and transcription factor family activity…”
Please write in full “LIHC, BLCA, BRCA, CHOL, KICH”. What is the real meaning of this information? Even if important, the authors might only hypothesize a correlation. Major criticisms in this point here must be inserted.
5. Conclusions
Page 14 The title indicating autophagy networks in this critical ending point are not described in detail. Which type of autophagy is involved (downregulated) and drives metabolic dysfunctions?
The authors might test autophagy modulators in this study because tuning autophagy alleviates liver damage. Detailed bioinformatic analysis has been properly used as a tool and interesting novel diagnostic opportunity but pathological questions must be better addressed. Have the authors an idea of autophagic flux? The extent and acceleration of the last step of macro autophagy is crucial for proper dismantling of cellular debris.
Author Response
Dear professor:
Thank you very much for your kindly comments on our manuscript titled "Transcriptome sequencing reveals autophagy networks in rat livers during the development of NAFLD and identifies autophagy hub genes ". We appreciate your time and effort in reviewing our work. Based on your suggestions, we carefully revised the manuscript. We are now sending the revised article for your re-consideration to publish in “Intemational Joural of Molecular Sciences”. Below you will find our point-by-point responses to the reviewers’ comments and questions, the original comments are in black, and our responses are in red. We believe that your comments have helped us to improve our manuscript and look forward to hearing from you soon for a favorable decision.
Thank you again for your time and consideration.
Yours sincerely,
Jian Xie
1. Abstract- Please the authors must rewrite abbreviations in full names for example ND38, HFD 38, PPI must be put inside brackets. Too many abbreviations make hard to understand the meaning for readers not so knowledgeable on the subject. If there are too many words i suggest inserting a generic term and describe details in the text (Materials and Methods). Genes firstly indicated here, must be written in full than abbreviated in brackets.
Answer: We sincerely appreciate the valuable comments. We are very sorry for the mistakes in the manuscript and inconvenience they caused in your reading. We have modified the abbreviations throughout the text to ensure that the full name is used upon first mention,and put the abbreviation in brackets.
2. Introduction-Page 2-Please delete “aging” the sentence must sound as “Autophagy can degrade dead organelles…” and insert proper References here.
Answer: Thank you for your feedback. We have deleted this sentence as required and added reference to support this statement.
3. Again, please rewrite in full genes indicated as Sox9, Ccl20, Cxcl1, Cd24---and also Nos3, Igf1, Vamp8, Fos, Hmox1.
Answer: We sincerely appreciate the valuable comments. We have given the full name of this genes in the manuscript.
4. GSE89632 abbreviation what means?
Answer: GSE89632 is abbreviated as the gene expression dataset "GSE89632" downloaded from the gene expression database (GEO), which includes 20 patients with simple nonalcoholic fatty liver, 19 patients with NASH and 24 healthy controls.
5. Please check the order of the words. The sentence at the end of Introduction must be rewritten as “Therefore, we constructed a high-fat diet-fed mouse model for short-term (16 weeks) or long-term ( 38 weeks), and induced NAFLD.”
Answer: We sincerely appreciate the valuable comments. We have changed this sentence to the correct expression according to your request as follow: “In this study, we established a high-fat diet mouse model to induce NAFLD in both short-term (16 weeks, HFD16) and long-term (38 weeks, HFD38) groups.”
6. Materials and Methods-Page 2-Please indicate the provider of different diets adopted in the study and the formulation of fat (coconut oil, soybean oil, lard) or a Reference indicating the information, which can be useful for other researchers interested in NAFLD. Different diet composition in fat even if provide similar calories might exert different effects in hepatic steatosis.
Answer: We sincerely appreciate the valuable comments. High fat diet (53% kcal from fat) was formulated according to previous publications [15]. The composition was displayed in Supplementary Table S1.
- Li, J.; Wu, H.; Liu, Y.; Yang, L. High fat diet induced obesity model using four strainsof mice: Kunming, C57BL/6, BALB/c and ICR. Exp Anim 2020, 69, 326-335, doi: 10.1538/expanim.19-0148.
7. Page 4-Indicate in full “TF analysis”
Answer: I'm sorry to bring you a bad reading experience. "TF" means "Transcription factors", and we have changed it to the full name in manuscript.
8. I suggest to estimate by image analysis program lipid droplets number inside H&E stained liver sections. This is important to reinforce microscopic feature of steatosis shown in Figure 2.
Answer: We sincerely appreciate the valuable comments. In this study we can clearly see that there are significant differences in lipid droplets between ND, HFD16weeks and HFD38weeks through HE staining, and we can get the results intuitively through pictures. Most of the existing literatures generally observe the tissue slices in pharmacodynamics through microscopy, and we will add relevant image data analysis in the following research.
9. Results. Page 8 Figure 4 must be enlarged because it is hard to understand . Please insert a brief explanation in Figure 4 legend.
Answer: Thank you for the comments. We have enlarged Figure 4 for better observation and reading.
10. Page 9 Please insert full names for Tfdp1, Srebf2, Irs2, Hoxa2,Pnpla2,Irs2. These abbreviations must be put in the list of abbreviations at the end of the text.
Answer: We sincerely appreciate the valuable comments. We have given the full names of all genes in manuscript and listed them in the abbreviation list at the end of the article.
11. Page 11 Please insert Figure 8 with more indications in Figure legend as Supplementary file (S1). There are too many data in the text and they are confounding.
Answer: We sincerely appreciate the valuable comments. We have added Figure legend to Figure 8 and introduced the results in detail in 3.8.
12. Discussion. Page 11-Please delete some sentences that have been just indicated in the Introduction and redundant. For example “At present, the diagnosis of NAFLD mainly depends on biochemical indicators, imaging and histopathological examination”.
Answer: We sincerely appreciate the valuable comments. We have deleted the following sentence from our discussion: "At present, the diagnosis of NAFLD mainly depends on biochemical indicators, imaging and histopathological examination."
13. Please the authors might indicate which type of autophagy has been defined by genic profile, ie non selective macro autophagy, or chaperone-mediated autophagy? Both mechanisms are deeply involved in the liver.
Answer: Thank you for the comments. The type of autophagy has been defined by genic profile is “chaperone-mediated autophagy”
14. Page 12 the relationship between colorectal cancer and NAFLD must be better discussed with appropriate References.
Answer: Thank you for the comments. We are discussing this issue as follows: “Researches has shown that NAFLD is a risk factor for colorectal cancer. The incidence of colorectal tumors in NAFLD patients is twice as high as that in the general population [21]. NAFLD may be an independent risk factor for the occurrence of colorectal cancer, and the more severe the NAFLD lesions, the higher the risk of colorectal cancer. It is recommended that NAFLD patients, especially those with more severe lesions, undergo regular colonoscopy screening to detect colorectal tumors early and reduce mortality rates [22,23].”
- Kim, K.W.; Kang, H.W.; Yoo, H.; Jun, Y.; Lee, H.J.; Im, J.P.; Kim, J.W.; Kim, J.S.; Koh, S.J.; Jung, Y.J. Association between severe hepatic steatosis examined by Fibroscan and the risk of high-risk colorectal neoplasia. PLoS One 2022, 17, e279242.
- Bray, F.; Ferlay, J.; Soerjomataram, I.; Siegel, R.L.; Torre, L.A.; Jemal, A. Global cancer statistics 2018: GLOBOCAN estimates of incidence and mortality worldwide for 36 cancers in 185 countries. CA Cancer J Clin 2018, 68, 394-424.
- Cho, Y.; Lim, S.K.; Joo, S.K.; Jeong, D.H.; Kim, J.H.; Bae, J.M.; Park, J.H.; Chang, M.S.; Lee, D.H.; Jung, Y.J.; Kim, B.G.; Kim, D.; Lee, K.L.; Kim, W. Nonalcoholic steatohepatitis is associated with a higher risk of advanced colorectal neoplasm. Liver Int. 2019, 39, 1722-1731.
- Page 13-Similarly, the late increase of Srebf2 at 38 weeks of HFD must be better discussed here with relative References.
15. Page 13-Similarly, the late increase of Srebf2 at 38 weeks of HFD must be better discussed here with relative References.
Answer: Thank you for your valuable feedback. We have re-discussed Srebf2 at 38 weeks of HFD as follows: “In this study, the cholesterol synthesis gene Srebf2 was significantly increased at 38 weeks with the prolongation of feeding on a high-fat diet. Srebf2 is a key regulator of cellular cholesterol homeostasis which encodes sterol regulatory element binding protein 2 (SREBP2) [28]. Its activation promotes hepatocyte cholesterol accumulation by coordi-nately activating cholesterol biosynthesis and uptake and repressing cholesterol excretion [29]. Srebf2 mRNA levels are significantly increased in patients with NASH, threefold higher than in non-NASH patients [30]. Cross sectional human studies found an in-creased hepatic expression of SREBP-2 and of its target genes, with the degree of SREBP-2 activation paralleling the severity of hepatic cholesterol overload and liver histology[31]. These results indicated that a long-term high-fat diet promoted liver lipid accumulation and aggravated NAFLD by upregulating cholesterol synthesis genes.”
- Kang, H.; You, H.J.; Lee, G.; Lee, S.H.; Yoo, T.; Choi, M.; Joo, S.K.; Park, J.H.; Chang, M.S.; Lee, D.H.; et al. Interaction effect between NAFLD severity and high carbohydrate diet on gut microbiome alteration and hepatic de novo lipogenesis. Gut Microbes 2022, 14, 2078612, doi: 10.1080/19490976.2022.2078612.
- Madison, B.B. Srebp2: A master regulator of sterol and fatty acid synthesis. J. Lipid Res. 2016, 57, 333-335, doi: 10.1194/jlr.C066712.
- Caballero, F.; Fernández, A.; De Lacy, A.M.; Fernández-Checa, J.C.; Caballería, J.; García-Ruiz, C. Enhanced free cholesterol, SREBP-2 and StAR expression in human NASH. J. Hepatol. 2009, 50, 789-796, doi: 10.1016/j.jhep.2008.12.016.
- Musso, G.; Cassader, M.; Bo, S.; De Michieli, F.; Gambino, R. Sterol regulatory element-binding factor 2 (SREBF-2) predicts 7-year NAFLD incidence and severity of liver disease and lipoprotein and glucose dysmetabolism. Diabetes 2013, 62, 1109-1120, doi: 10.2337/db12-0858.
16. Page 14- the English txt is understandable. Please rewrite the sentence “ The gene expression protein has DNA bi and transcription factor family activity…”
Answer: We sincerely appreciate the valuable comments. We have rewritten this sentence as “The protein encoded by the Hoxa2 possesses both DNA binding and transcription factor activities. It plays a crucial role in regulating the proliferation and differentiation of embryonic and adult cells by binding to specific sites on DNA and initiating the transcription of target genes.”
17. Please write in full “LIHC, BLCA, BRCA, CHOL, KICH”. What is the real meaning of this information? Even if important, the authors might only hypothesize a correlation. Major criticisms in this point here must be inserted.
Answer: I'm sorry to bring you a bad reading experience. We have modified the abbreviations and given their full names in the manuscript.
18. Conclusions. Page 14 The title indicating autophagy networks in this critical ending point are not described in detail. Which type of autophagy is involved (downregulated) and drives metabolic dysfunctions?
Answer: Thank you for the comments. Upregulating the cholesterol synthesis gene Srebf2 may exacerbate NAFLD by promoting hepatic lipid accumulation. Conversely, downregulating the fatty acid oxidation genes Pnpla2 and Plin2 may promote the formation of intracellular lipid droplets, while downregulating the Irs2 gene may induce insulin resistance and inhibit autophagy in liver cells, ultimately triggering NAFLD.
19. The authors might test autophagy modulators in this study because tuning autophagy alleviates liver damage. Detailed bioinformatic analysis has been properly used as a tool and interesting novel diagnostic opportunity but pathological questions must be better addressed. Have the authors an idea of autophagic flux? The extent and acceleration of the last step of macro autophagy is crucial for proper dismantling of cellular debris.
Answer: Thank you for the comments. Autophagy is a self-protective mechanism widely present in eukaryotic cells, which can maintain cellular homeostasis by renewing cellular metabolism and energy through self-digestion. Autophagic flux refers to the rate and efficiency of the autophagic process, which represents the cell's ability to clear waste. It is measured by changes in autophagic enzyme activity. Macroautophagy is the final step of autophagy, also known as the formation of autolysosomes. During macroautophagy, autophagosomes fuse with lysosomes to form autolysosomes, where enzymes can break down waste into small molecules for the cell to reuse. Decreased autophagic flux or inhibited macroautophagy can result in the accumulation of waste inside the cell, which may cause damage to the cell's survival and function, and even lead to cell death. Accelerating autophagic flux and promoting the final step of macroautophagy is crucial for proper degradation of cellular waste. This can be achieved through various means, including the use of drugs, nutrition, and exercise to promote autophagy.
Reviewer 3 Report
The manuscript, “Transcriptome sequencing reveals autophagy networks in rat livers during the development of NAFLD and identifies autophagy hub genes” by Xie et. al., analyzed gene expressions in NAFLD mouse model and identified a group of potential genes that may be
Used as key therapeutic targets and early diagnostic markers in the progression in the future. The detailed analysis and the status of the gene expression during NAFLD will be of great interest in the relevant fields. However, I have following concerns:
1. The data presented show both up- and down-regulation of autophagy genes. Upregulated ones were, as mentioned, associated with autophagy positive regulation of autophagy, process utilizing autophagy. The downregulated DEGs also included macroautophagy, CMA, positive regulation of autophagy. It is difficult to understand how they are useful as future targets for NAFLD or as diagnostic markers if they go in both directions.
2. Only male mice were used in the analysis. Although it is thought that women get less NFALD than women, it is still a significant portion in older women.
3. In Figure 2, please add a panel of either Sirius Red or Trichrome.
4. Detail legend for Figure 4. Figure 5 is missing.
5. No validation by qPCR or western blots for the 31 genes. Please validate some of the up-and-down-regulated genes using the conventional method.
6. The expression states of significant autophagy genes, such as ATG3, ATG5 LC3, TFEB, p62, etc. are not detailed. Please mention your findings of these genes.
7. Please present an analysis of the mTOR pathway from the gene expression data.
8. Please rewrite identical phrases that were taken from Kung et al., from Gut Microbes, 2022 J
Author Response
Dear professor:
Thank you very much for your kindly comments on our manuscript titled "Transcriptome sequencing reveals autophagy networks in rat livers during the development of NAFLD and identifies autophagy hub genes ". We appreciate your time and effort in reviewing our work. Based on your suggestions, we carefully revised the manuscript. We are now sending the revised article for your re-consideration to publish in “Intemational Joural of Molecular Sciences”. Below you will find our point-by-point responses to the reviewers’ comments and questions, the original comments are in black, and our responses are in red. We believe that your comments have helped us to improve our manuscript and look forward to hearing from you soon for a favorable decision.
Thank you again for your time and consideration.
Yours sincerely,
Jian Xie
reviewer # 3:
1. The data presented show both up- and down-regulation of autophagy genes. Upregulated ones were, as mentioned, associated with autophagy positive regulation of autophagy, process utilizing autophagy. The downregulated DEGs also included macroautophagy, CMA, positive regulation of autophagy. It is difficult to understand how they are useful as future targets for NAFLD or as diagnostic markers if they go in both directions.
Answer: Thank you for the comments. Autophagy is an intracellular process that can be regulated through multiple pathways. Autophagy is believed to play an important role in the development and progression of NAFLD. Our results indicate that Srebf2 is upregulated, while Pnpla2, Plin2, and Irs2 are downregulated, suggesting that the autophagy pathway is bidirectionally regulated. Upregulating the cholesterol synthesis gene Srebf2 may exacerbate NAFLD by promoting hepatic lipid accumulation. Conversely, downregulating the fatty acid oxidation genes Pnpla2 and Plin2 may promote the formation of intracellular lipid droplets, while downregulating the Irs2 gene may induce insulin resistance and inhibit autophagy in liver cells, ultimately triggering NAFLD.
Autophagy involves multiple branches and regulatory mechanisms. Upregulated and downregulated genes may play a role in different branches or mechanisms. In addition, the regulation of autophagy pathway may depend on different conditions, such as cell type and environmental pressure. Therefore, further research is needed to understand the specific functions of these genes and how to play a role in NAFLD.
Finally, whether these genes can be used as future targets or diagnostic markers of NAFLD needs further study. The genes with bidirectional regulation may have different functions and affect NAFLD in different ways. Therefore, it is necessary to study the function and regulation mechanism of these genes in order to better understand their role in NAFLD and determine whether they have potential therapeutic or diagnostic value.
- Madison, B.B. Srebp2: A master regulator of sterol and fatty acid synthesis. J. Lipid Res. 2016, 57, 333-335
- Caballero, F.; Fernández, A.; De Lacy, A.M.; Fernández-Checa, J.C.; Caballería, J.; García-Ruiz, C. Enhanced free cholesterol, SREBP-2 and StAR expression in human NASH. J. Hepatol. 2009, 50, 789-796
- Sanchez-Lazo, L.; Brisard, D.; Elis, S.; Maillard, V.; Uzbekov, R.; Labas, V.; Desmarchais, A.; Papillier, P.; Monget, P.; Uzbekova, S. Fatty Acid Synthesis and Oxidation in Cumulus Cells Support Oocyte Maturation in Bovine. Mol. Endocrinol. 2014, 28, 1502-1521
- Taxiarchis, A.; Mahdessian, H.; Silveira, A.; Fisher, R.M.; Van'T Hooft, F.M. PNPLA2 influences secretion of triglyceride-rich lipoproteins by human hepatoma cells. J. Lipid Res. 2019, 60, 1069-1077
- Zhou, J.; Zhang, N.; Aldhahrani, A.; Soliman, M.M.; Zhang, L.; Zhou, F. Puerarin ameliorates nonalcoholic fatty liver in rats by regulating hepatic lipid accumulation, oxidative stress, and inflammation. Frontiers in Immunology 2022, 13
- Mcmanaman, J.L.; Bales, E.S.; Orlicky, D.J.; Jackman, M.; Maclean, P.S.; Cain, S.; Crunk, A.E.; Mansur, A.; Graham, C.E.; Bowman, T.A.; et al. Perilipin-2-null mice are protected against diet-induced obesity, adipose inflammation, and fatty liver disease. J. Lipid Res. 2013, 54, 1346-1359
- Jin, Y.; Tan, Y.; Chen, L.; Liu, Y.; Ren, Z. Reactive Oxygen Species Induces Lipid Droplet Accumulation in HepG2 Cells by Increasing Perilipin 2 Expression. Int. J. Mol. Sci. 2018, 19, 3445
- Tsai, T.H.; Chen, E.; Li, L.; Saha, P.; Lee, H.J.; Huang, L.S.; Shelness, G.S.; Chan, L.; Chang, B.H. The constitutive lipid droplet protein PLIN2 regulates autophagy in liver. Autophagy 2017, 13, 1130-1144
- Yan, R.; Niu, C.; Tian, Y. Roles of Autophagy and Protein Kinase C-epsilon in Lipid Metabolism of Nonalcoholic Fatty Liver Cell Models. Arch. Med. Res. 2018, 49, 381-390
2. Only male mice were used in the analysis. Although it is thought that women get less NFALD than women, it is still a significant portion in older women.
Answer: Thank you for the comments. The reason why we choose to use only male mice is that studies show that male mice are more likely to develop NAFLD, and the influence of estrogen may lead to the uncertainty of data. In addition, we also recognize that women may also suffer from NAFLD, and we will explore the influence of gender differences in future research. Thank you again for your question.
3. In Figure 2, please add a panel of either Sirius Red or Trichrome.
Answer: We sincerely appreciate the valuable comments. NAFLD is a common liver disease that usually includes hepatic steatosis and nonalcoholic steatohepatitis, with the main pathological features being liver fat accumulation and inflammation. Pulmonary fibrosis is a pathological change associated with lung disease, which refers to fibrosis and scarring of lung tissue. Although NAFLD and pulmonary fibrosis are both associated with metabolic disorders, there is no necessary connection between them. Not all NAFLD patients will have pulmonary fibrosis. HE staining can be used to observe the histological morphology of the liver and changes in liver cells, including steatosis, inflammatory cell infiltration, and cell death, among others. Therefore, it is usually not necessary to use Sirius Red or Trichrome to observe liver fibrosis. We will also consider your opinions and use multiple methods to study the pathological manifestations of NAFLD in future research.
4. Detail legend for Figure 4. Figure 5 is missing.
Answer: We sincerely appreciate the valuable comments. We are very sorry for the mistakes in the manuscript and inconvenience they caused in your reading. We have added the newly detail legend for Figure 4 and Figure 5 as follow:
Figure 4. Functional enrichment analysis. (A) GO enrichment analyses of 31 common DEGs. The top 10 terms in each GO category (MF,molecular function; CC, cellular components; BP, biological processes). Red represents up-regulated genes and blue represents down-regulated genesï¼›(B) KEGG enrichment analyses of 31 common DEGs. All significant KEGG pathways. The cut-off standard is FDR<0.05. The color represents the FDR of this enrichment pathway, and the red represents a more prominent FDR. Gene ratio refers to the proportion of genes enriched in each pathway. Triangles represent upregulated genes, and dots represent downregulated genes.
Figure 5. Screening of protein interactions (PPIs) and hub genes. (A) PPI network of 31 DEGs. The minimum required interaction score is 0.4, consisting of 25 nodes (genes) and 36 edges (interactions); (B) Screening four hub genes based on seven algorithms; (C) Hub gene Irs2, Pnpla2, Srebf2, and Plin2 expression. The expression of Irs2, Pnpla2, and Plin2 was significantly downregulated in the 38 week high fat diet, while the expression of Srebf2 was significantly upregulated in the 38 week high fat diet. * p < 0.05; ** p < 0.01; ***p < 0.001; **** p < 0.0001; ns p > 0.05.
5. No validation by qPCR or western blots for the 31 genes. Please validate some of the up-and-down-regulated genes using the conventional method.
Answer: Thanks for your great suggestion. It would be perfect once the data was validated by qPCR or WB for some of the investigated genes. It’s just in our schedule when we do the next animal experiment. However, we have to please you understand that the samples have already been out of use because the mouse livers were too small to support various assays. It will need more than 1 year to finish the next animal experiment for the liver tissues. Fortunately, we have checked the transcriptome data in another paper use the same dataset (He, Yuqi et al. “High fat diet significantly changed the global gene expression profile involved in hepatic drug metabolism and pharmacokinetic system in mice.” Nutrition & metabolism vol. 17 37. 24 May. 2020, doi:10.1186/s12986-020-00456-w). Although genes in that paper were related to drug metabolism, we hope it could be used as one of the evidence that the transcriptome data is reliable as they came from the same tissues and same sequencing process as the data in our current manuscript. Moreover, we have replicates in the transcriptome data, and all conclusions would be validated later in our mechanism study. Thanks for your kindly understandings.
6. The expression states of significant autophagy genes, such as ATG3, ATG5 LC3, TFEB, p62, etc. are not detailed. Please mention your findings of these genes.
Answer: Thank you for your valuable feedback. We appreciate your concerns regarding the expression states of significant autophagy genes, such as ATG3, ATG5 LC3, TFEB, p62, etc. in our study. As you rightly pointed out, ATG3, ATG5 LC3, TFEB, and P62 are important autophagy genes. However, we would like to clarify that we screened the differential genes according to FDR<0.05, |log2 (multiple change) |≥1, and these genes were not significantly different only by using FPKM values in our transcriptome data. We acknowledge the importance of these genes in the regulation of autophagy and appreciate your suggestion to provide more details about them. However, as we did not observe any significant differential expression of these genes in our study, we believe that including their expression data in our manuscript would not add value to the study and could potentially be misleading.
7. Please present an analysis of the mTOR pathway from the gene expression data.
Answer: Thank you for the comments. mTOR is the downstream target of PI3K/Akt signaling pathway that plays a critical role in regulating cellular growth, metabolism, and autophagy. The activation of mTOR in NAFLD has several downstream effects. It leads to increased lipid synthesis, which exacerbates the accumulation of fat in the liver. Additionally, mTOR activation suppresses autophagy, which is the process by which cells recycle damaged or excess cellular components. Impaired autophagy contributes to the accumulation of damaged proteins and organelles in the liver, further exacerbating the disease's progression.
8. Please rewrite identical phrases that were taken from Kung et al., from Gut Microbes, 2022 J
Answer: Thank you for your valuable feedback. We have rewrite the discussion about Srebf2 as follows: “In this study, the cholesterol synthesis gene Srebf2 was significantly increased at 38 weeks with the prolongation of feeding on a high-fat diet. Srebf2 is a key regulator of cellular cholesterol homeostasis which encodes sterol regulatory element binding protein 2 (SREBP2) [28]. Its activation promotes hepatocyte cholesterol accumulation by coordi-nately activating cholesterol biosynthesis and uptake and repressing cholesterol excretion [29]. Srebf2 mRNA levels are significantly increased in patients with NASH, threefold higher than in non-NASH patients [30]. Cross sectional human studies found an in-creased hepatic expression of SREBP2 and of its target genes, with the degree of SREBP2 activation paralleling the severity of hepatic cholesterol overload and liver histology[31]. These results indicated that a long-term high-fat diet promoted liver lipid accumulation and aggravated NAFLD by upregulating cholesterol synthesis genes.”
Round 2
Reviewer 2 Report
This version has been ameliorated the authors answered properly to the majority of reviewer's criticisms. The strength of this study is detailed bioinformatic gene analysis, the weakness the loss of detailed histopathological/ultrastructural changes in the liver. These two different aspects of NAFLD might be better correlated. However, in my opinion, the study might be accepted in IJMS.
Author Response
Dear professor:
Thank you very much for your kindly comments on our manuscript titled "Transcriptome sequencing reveals autophagy networks in rat livers during the development of NAFLD and identifies autophagy hub genes ". We appreciate your time and effort in reviewing our work. Based on your suggestions, we carefully revised the manuscript. We are now sending the revised article for your re-consideration to publish in “Intemational Joural of Molecular Sciences”. Below you will find our point-by-point responses to the reviewers’ comments and questions, the original comments are in black, and our responses are in red. We believe that your comments have helped us to improve our manuscript and look forward to hearing from you soon for a favorable decision.
Thank you again for your time and consideration.
Yours sincerely,
Jian Xie
1. This version has been ameliorated the authors answered properly to the majority of reviewer's criticisms. The strength of this study is detailed bioinformatic gene analysis, the weakness the loss of detailed histopathological/ultrastructural changes in the liver. These two different aspects of NAFLD might be better correlated. However, in my opinion, the study might be accepted in IJMS.
Answer: Thank you very much for your constructive comments and suggestions on our manuscript. Although our manuscript still has some limitations, we appreciate your insightful feedback, which will help us to conduct further in-depth research in the future. We are grateful for your valuable input and would like to express our sincere thanks for your constructive review. We hope to meet the publishing requirements of IJMS and look forward to hearing back from you soon.
Reviewer 3 Report
The authors have adequately answered and provided explanations for not presenting more data. Although more data would have increased the quality, it is still a good manuscript.
Author Response
Dear professor:
Thank you very much for your kindly comments on our manuscript titled "Transcriptome sequencing reveals autophagy networks in rat livers during the development of NAFLD and identifies autophagy hub genes ". We appreciate your time and effort in reviewing our work. Based on your suggestions, we carefully revised the manuscript. We are now sending the revised article for your re-consideration to publish in “Intemational Joural of Molecular Sciences”. Below you will find our point-by-point responses to the reviewers’ comments and questions, the original comments are in black, and our responses are in red. We believe that your comments have helped us to improve our manuscript and look forward to hearing from you soon for a favorable decision.
Thank you again for your time and consideration.
Yours sincerely,
Jian Xie
1. The authors have adequately answered and provided explanations for not presenting more data. Although more data would have increased the quality, it is still a good manuscript.
Answer: Thank you for your thorough review of our manuscript. We greatly appreciate your recognition of our work, and we agree that there is room for improvement in our research. However, we believe that the strength of our study lies in the analysis of bioinformatics data, and we plan to conduct more extensive and in-depth research in our next steps. We are grateful for your valuable comments and constructive feedback. We will take all of your suggestions into consideration in our revisions. Our goal is to meet the publication standards of IJMS, and we look forward to hearing back from you soon. Thank you again for your time and effort in reviewing our manuscript.